# ESIB's Antecedents: An Analytic Hierarchy Process Application in the Manufacturing Industry in Albania

Ardita Malaj [1,*], Selim Zaim [2], Nizamettin Bayyurt [1] and Merve Tarim [3]

1   Management Faculty, Istanbul Technical University, Istanbul 34485, Turkey; bayyurt@itu.edu.tr
2   Management Faculty, Ibn Haldun University, Istanbul 34480, Turkey; selim.zaim@ihu.edu.tr
3   Engineering Faculty, Halic University, Istanbul 34381, Turkey; mervetarim@halic.edu.tr
*   Correspondence: malaj19@itu.edu.tr

**Abstract:** This study examined factors that might motivate employees to engage in social innovation. The objectives of this study were to identify the relative importance of factors that impact Employee Social Intrapreneurial Behavior (ESIB) in the manufacturing sector in Albania using an AHP approach and to select the best practice that can improve Employee Social Intrapreneurial Behavior (ESIB) on a company basis in the same sector by merging a linguistic method with the AHP approach. A questionnaire was designed to collect expert viewpoints. A standardized AHP scale ranging from 1 to 9 was used in the questionnaire. Fifteen managers and experts from four important manufacturing companies in Albania (Everest Shpk., Kamëz, Albania; Lufra Shpk., Lushnje, Albania; Ajka Shpk., Lushnje, Albania and F&M Shpk., Tirane, Albania) were contacted, and eleven of them expressed an interest in engaging in our study by ranking the importance of various criteria and sub-criteria. The findings suggest that the sub-components of the internal factor, such as superior relationship quality and expected image gains, have a combined importance of 50% in the local weights in the Albanian context. The two sub-components mentioned above remain significant in the analysis of global weights, accounting for 41% of global influence. These findings confirm Hofstede's evaluation of the Albanian culture, where the power distance dimension ranks first, accounting for 90% of the variance and thus classifying the society as very hierarchical. In such a cultural context, people accept and try to please their leaders. Furthermore, high uncertainty avoidance determines individual motivation for financial gains in an unsecure economic environment. We proposed and examined two programs that businesses could implement to boost employee participation in innovative behaviors, and the results suggest that organizations should implement a program called the Workshop. The core idea of this program is to provide monthly workshops focusing on improving humorous situations that may have positive effects on ESIB. Due to the small number of participants in this study, a future study might want to examine the viewpoints of employees as well as other stakeholders, such as NGOs or legal institutions.

**Keywords:** ESIB; internal factor; AHP; manufacturing; Albania





## 1. Introduction

Employee behavior in the workplace has always been an intriguing topic for researchers. There is also research examining the role of employees' innovative behavior. Fostering social innovation is becoming a major priority. This may be accomplished using social innovation practices and behaviors that can be generated for consumer products that do not harm the community or the environment. Our main objective is to propose an innovative concept identified as Employee Social Intrapreneurial Behavior (ESIB), which focuses on employees' innovative behaviors. We aim to examine some of the antecedents of ESIB in the Albanian manufacturing industry. Our findings can provide essential information that helps organizations establish specific practices to foster innovative behaviors among employees.

Over time, the role of employees in businesses has evolved, granting them more autonomy, responsibility, and involvement in decentralized decision-making processes [1]. Employee intrapreneurial behaviors, often described as a bottom-up or behavioral approach to driving change and improvement, have gained attention [2]. Taking a behavior-based approach complements the organizational-level perspective and offers a more comprehensive understanding of intrapreneurship [3]. Regarding organizational entrepreneurship, it is acknowledged that Gifford Pinchot coined the word "intrapreneur" in 1978 [4]. The authors of [5] described intrapreneurship as a radical concept that harnesses employees' entrepreneurial abilities to drive innovation within organizations. In [6], it is defined as the collective effort of all employees to engage in new business activities within their organization. Similarly, the authors of [7] characterized internal entrepreneurship as the capacity to innovate, take risks, and proactively compete. The authors of [8] further elaborated on intrapreneurial activities as daily innovations that enhance an organization's responsiveness to clients, whether through ideas, a drive for self-transcendence, or cost-effective service provision. The authors of [9] emphasized the idea of employees occupying lower positions in the organizational hierarchy but having the autonomy to generate novel ideas, think creatively to address complex problems, and drive change within the organization. We believe that introducing a new concept of innovative behavior, Employee Social Intrapreneurial Behavior, will contribute to the literature and take it to a new level. The economic history of developing countries highlights the crucial role that industries play in driving and supporting their economic development. Industrial development serves as a primary catalyst and carrier for these countries' progress. One advantage of industrial development is an inherent need for continuous changes in industrial structures. This aspect is particularly significant for countries characterized by low production capacity, limited export volume, and inadequate levels of industrialization. In this context, manufacturing level affects the achieved degree of development of the whole country; it dictates the level of productivity and competitiveness of the industry [10].

Our study makes three contributions to the research field: First, we propose a new concept called Employee Social Intrapreneurial Behavior that centers around employee behavior. Second, there is a scarcity of research of this type that uses the AHP to analyze employees' social innovative behaviors and classify the influencing factors into two major groups: internal organizational context and external organizational context. Third, this is the first paper examining these factors in the Albanian context, and it will be very useful for managers and HR to adopt some of the recommendations in our paper in the workplace.

This paper is divided into the following sections: Section 2 discusses in detail the definition of ESIB and factors that impact it. Section 3 presents a review of factors that influence employee innovative behaviors in the manufacturing industry. Section 4 illustrates the AHP approach. Section 5 covers the AHP application conducted in this study. Sections 6 and 7 present the findings, discuss the conclusions, and offer recommendations for further studies.

## 2. Definition of Employee Social Intrapreneurial Behavior (ESIB)

In the history of entrepreneurship, Schumpeter's entrepreneurship appears to mark an important moment [11]: "Schumpeter emphasized the role of the innovator-to-be, the innovator, the developer, the promoter, the person who initiatives and recognizes technical improvements and who succeeds in getting them introduced" [12]. In this circular flow, an entrepreneur is the only agent of economic change. In some ways, an entrepreneur personifies innovation as a person who carries out new combinations. However, Schumpeter pointed out that in the world of large companies, an entrepreneur is not always an independent economic agent but can also be an employee of a large corporation with an entrepreneurial function. Here, the role of entrepreneurial skills is emphasized once again but, this time, regarding the value of cooperative entrepreneurship in large corporations, rather than the "heroic" creative labor of a single entrepreneur.

Thus, intrapreneurship focuses on individual-level behaviors. The significance of employee intrapreneurial behavior (IB) in achieving organizational success has been widely acknowledged [13]. Employees are expected to be active participants in their roles, taking on the roles of "innovators" and "differentiators" rather than passively accepting changing tasks and products [14]. Intrapreneurial behaviors can be conceptualized in various ways. However, extant definitions do not specify the unique intrapreneurial aspects of these behaviors or their contributions to entrepreneurial outcomes. While intrapreneurship is acknowledged as a broad concept that encompasses intrapreneurial and proactiveness, there is less consensus on its precise definition. Several academics have defined intrapreneurship as bottom-up, proactive work-related behaviors of individual employees with the ability to transform ideas into commercial success [15].

Social intrapreneurship represents a unique amalgamation of concepts encompassing social responsibility, internal dynamics, and entrepreneurship. Firstly, it is driven by a commitment to social responsibility, aiming to address social or environmental issues that extend beyond the interests of individuals or private organizations [16]. Secondly, it operates within the framework of existing organizations, emphasizing how social intrapreneurial initiatives can align with core organizational objectives and shape long-term direction. Lastly, it follows an entrepreneurial process that involves identifying and exploring opportunities to generate future goods and services [17]. This distinctive blend of social responsibility, organizational context, and entrepreneurship is spearheaded by employees and is characterized by its voluntary and non-formalized nature.

Social intrapreneurship can be considered as a subcategory of corporate social responsibility (CSR) as it involves enhancing social responsibility within established firms [16]. It goes beyond broader CSR by emphasizing entrepreneurial action and innovation as key drivers. To pursue projects with social and environmental goals while considering the financial obligations of organizations, one must practice social intrapreneurship.

Social intrapreneurship, as opposed to corporate social responsibility, comprises the discovery of chances to develop new products or services that address social or environmental problems [18]. The focus is on discovering new means–ends relationships to solve social and environmental issues, rather than solely optimizing existing means–ends frameworks [17].

The interpretation of employee innovative behavior varies, with some studies presenting it as a unified concept [19], while others provide extensive lists that include up to 16 distinct characteristics [20]. However, the more comprehensive lists encompass various qualitative elements, such as personality traits (e.g., risk taking), specific behaviors (e.g., internal coalition building), and clusters of behaviors (e.g., championing). Through the analysis of relevant scales and the literature, eight fundamental aspects of innovative behavior that consistently contribute to the generation of innovative outputs have been identified. We propose a new concept named Employee Social Intrapreneurial Behavior (ESIB), which includes both intrapreneurial and socially focused behavior.

The following describes Employee Social Intrapreneurial Behavior (ESIB):

> *"It is a behavior displayed by personnel employed within an organization, regardless of the type of organization. Employees feel appreciated and a part of the organization's success. They generate societal ideas for new procedures, services, or products."*

The main idea of ESIB is that innovation comes from the bottom of an organization; employees engage in thinking, proposing, and implementing innovative ideas while keeping the social aspect in mind; as a result, they can come up with new ways of doing the work and procedures and new product/service ideas that will benefit the organization and society while not harming it.

We discovered a significant gap in the literature while screening previous studies that assess concepts such as innovation, entrepreneurship, social innovation, and employee intrapreneurial behavior using the AHP as a method. The AHP technique has been applied in very few research papers that address employee innovative behaviors. The study [21]

applied this technique. In their work, they examined the factors that motivate employees' creative behaviors.

Introducing a novel concept, such as Employee Social Intrapreneurial (ESIB), can add to the literature while also utilizing the AHP as a method.

Albania has been struggling to create and maintain a strong industrial sector due to the small size of the country's economy. Following the tumultuous period of 1997–1998, Albania successfully fostered a favorable economic environment by facilitating the emergence of small workshops. These workshops served as the foundation for the reindustrialization process, laying the groundwork for further economic growth and development. Industry (including construction) remains a weak sector for the economy of Albania, with the value it adds to the GDP not exceeding USD 3 billion. Manufacturing output is still very low, and it even dropped by 6.08 percent in the first quarter of 2023 compared to the same period in the previous year [22]. Considering the recent reindustrialization process and the weakness of the manufacturing sector in Albania, this study focused specifically on this sector, so to genuinely contribute to the increase in social innovation capacities, the end result of which will be a better overall performance.

### 3. ESIB: External and Internal Factors

In this study, factors that affect ESIB were divided into two categories. The first group comprises external aspects that affect ESIB indirectly, whereas the second group consists of internal components that directly affect ESIB. The classification was based on previous research. External factors are those that are related to the external environment, the social background of the organization, or the country. After reviewing previous studies by various authors, we chose five factors that showed significant value in past studies. Our paper focuses on the manufacturing sector since the companies in this area have a big employee structure and higher potential for social innovation.

According to the literature, the following are the top five external factors that indirectly affect ESIB:

- Search breadth [23]: This factor relates to the capacity for learning from various partners through first-hand experience. It includes the ability to acquire and assimilate resources from many different places, including customers, suppliers, rivals, universities or other knowledge institutions, and the public sector or government. Therefore, it is anticipated that a wider search scope will lead to more developed acquisition and assimilation capabilities.
- Collectivism culture [24]: This factor pertains to a psychological predisposition that values group interests over those of the individual. It highlights the significance of a cultural orientation that values and emphasizes collective well-being and goals.
- Dynamic work environment [25]: This factor pertains to the speed, complexity, and nature of environmental changes in the work context. It acknowledges that innovation can be influenced by the dynamic nature of the work environment.
- Corporate reputation [26]: This factor refers to the general perceptions about a company's capacity and willingness to satisfy the demands of diverse stakeholders. A positive corporate reputation can influence and enhance the propensity for innovative behavior.
- Culture intelligence [27]: This factor reflects an individual's adaptability in an intercultural environment. It recognizes the importance of being able to navigate and understand different cultures, which can contribute to innovative behavior. Top of Form.

Effective alliance procedures give organizations access to outside expertise, which eventually improves performance, in accordance with the tenets of the dynamic capability theory [28]. The authors of [29] emphasized the value of conducting extensive external knowledge searches that involve partnerships with academic institutions, rival businesses, and suppliers. For a variety of reasons, organizations use the knowledge they receive from

outside partners. The ability of small- and medium-sized enterprises (SMEs) to absorb external knowledge is improved by the breadth of their search, which refers to the diversity of their external innovation partners [30]. According to [23], collaboration between internal and external parties encourages the growth of dynamic knowledge competency, which in turn produces competitive advantage through service innovation.

In [31], collectivism is defined as a psychological trait reflecting an individual's level of concern for others and the collective. It entails conforming to role norms. A previous study [32] showed that collectivism plays a crucial role in examining cultural influences, particularly in the context of Chinese studies. The authors found that collectivism acts as a positive moderator in the relationship between information sharing and innovative behavior, and it acts as a significant cultural factor that influences how sharing of knowledge affects creative behavior.

The ability of an employer to foresee events and their effects on the organization is hampered by the uncertainty present in the dynamics of the business environment [33]. Addressing the most recent requirements and changes is essential for an organization's survival and expansion in a dynamic environment. Both the organization and its personnel are greatly affected by a dynamic work environment. When faced with intricate, ambiguous, and tumultuous environmental changes, managers' and employees' behaviors are influenced by a certain degree of uncertainty [34]. Previous research on the relationship between environmental unpredictability and performance has primarily focused on organizational research or research at the management team level [35]. However, organizational dynamics poses challenges not just for businesses and their executives, but also for workers. Rapid environmental changes put employees under more stress at work; thus, it is important to have a depth of knowledge to lessen the effects of uncertainty [36]. According to [37], employees perform more innovatively when they operate in a dynamic workplace. In a fast-paced workplace, having exceptional workers is essential to fostering organizational success.

Social identity theory emphasizes an individual's sense of belonging to different social groups and suggests that social identity helps individuals define and establish their place in society [38]. According to [39], when employees identify with their organization, they are more likely to support it by participating in behaviors that benefit their organization. Corporate reputation is recognized as a significant managerial objective and can foster increased employee engagement [40]. In this regard, employee engagement within the organization is a vital prerequisite for innovative performance [41]. The authors of [42] suggested that the indirect relationship between corporate reputation and innovative job performance is supported by organizational engagement. Although corporate reputation may not directly impact innovative job performance, it demonstrates an indirect effect that is mediated through organizational engagement.

Studies that treat cultural intelligence (CQ) as a precursor have linked it to cross-cultural adjustment, judgment, and expatriate performance. Longitudinal studies have been carried out to pinpoint elements that improve cross-cultural interactions [43].

According to [44], behavioral CQ is the ability to adapt verbal and nonverbal acts to fit into a multicultural environment.

The concept of cultural intelligence is an extension of general intelligence that emphasizes a person's capacity for efficient intercultural communication [45]. According to [46], it is conceptualized as a complex construct encompassing elements of metacognition, cognition, motivation, and behavior. Individuals with high CQ possess the capacity to adapt to cross-cultural situations by effectively managing unique tasks and finding innovative solutions to old challenges [47].

High CQ can foster innovative behavior among multicultural personnel due to its favorable effects on cognitive flexibility [48]. Innovative thoughts and actions of employees are the product of social interaction among team members as well as individual thoughts. As a result, it is essential that different and unique ideas and knowledge be shared freely and openly for individual innovation [49]. This viewpoint holds that effective teamwork

supports an individual's innovative ideas and actions. Effective engagement becomes essential for learning information and valuable resources from one another in a multicultural workplace with employees from diverse cultures. Employees with higher CQ demonstrate greater motivation to interact often and productively with coworkers from various cultural backgrounds, which can enhance their social context centrality and allow them to learn a variety of skills from others [50].

When dealing with new, risky, and difficult jobs, employees' confidence can be increased and negative emotions can be decreased with appropriate support and encouragement from other team members [51]. As a result, workers are more likely to engage in creative jobs, exert significant effort to accomplish difficult goals, and develop and implement innovative ideas even in trying situations [52]. A longitudinal pilot study [53] provided additional evidence that cultural intelligence (CQ) training can support the growth of unique and innovative work-related behaviors.

Internal factors are elements that relate to human nature in taking decisions and can be critical when it comes to social intrapreneurial behavior. In addition, for the five internal factors chosen, we reviewed previous studies and selected the most important factors with a high level of significance in this context.

*The following are the top five internal factors that directly affect ESIB*:

- Perceived organizational support [54]: This factor refers to the extent to which employees perceive the organization supports them in terms of their work and overall well-being.
- Expected image gains [55]: This factor relates to employees' motivation to innovate based on the expectation of enhancing their positive image within the organization, thus promoting self-enhancement.
- Need for cognition [56]: This factor represents individuals' dispositional tendency to engage in and enjoy thinking. It reflects the intrinsic motivation of individuals to engage in cognitive activities, which can contribute to innovative behavior.
- Superior relationship quality [57]: This factor pertains to how superiors build relationships with their subordinates and create in-group and out-group dynamics within their work units. The quality of the relationship between superiors and subordinates can influence employees' innovative behavior.
- Perceived deviance tolerance [58]: This factor encompasses employees' perception of their leaders' tolerance of deviant behavior. It reflects the extent to which leaders are perceived as accepting or tolerant of deviations from norms, which can impact employees' willingness to engage in innovative behavior.

The organizational environment serves as a crucial contextual component that communicates behavioral expectations and potential outcomes [59]. When an organization's norms prioritize change over tradition, its members are more likely to initiate culturally appropriate changes. The authors of [60] highlighted that if employees perceive that their organization fosters innovation, they are motivated to engage in innovative behavior as a form of reciprocation for the company's support. In [61], a positive association between perceived organizational support and work engagement was identified, showing that employees would contribute more to their organization by demonstrating innovative behaviors when there was higher perceived organizational support.

In the research conducted by the authors of [54], it was also found that perceived organizational support had a significant positive impact on innovative behaviors.

More recently, innovation research has increasingly focused on social and political processes, providing insights into how innovation is implemented in the real world, rather than solely considering how it should be done [62]. Studies have indicated that image or legitimacy factors play a significant role in explaining decisions related to the adoption of innovation [63]. In a similar vein, the authors of [64] put forth a theoretical framework that contends that the innovation process can be seen as a "fad" or "fashion" in which innovations are occasionally adopted for their symbolic value, such as signaling innovation, rather

than for enhancing organizations' economic performance. This alternate viewpoint, which emphasizes the symbolic value of innovative actions and the significance of image considerations in innovation decisions that go beyond straightforward efficiency calculations, is known as the social-political perspective [55]. However, in their research, the authors of [55] found that expected image gains had a significant negative impact on innovative behaviors. They suggested one possible explanation is that individuals who intend to use innovative behaviors to impress or please others may be perceived as less innovative by their managers, resulting in a negative judgment of their innovative behaviors.

The research conducted by the authors of [65] indicated that the need for cognition is positively associated with individual innovative behaviors. They found empirical evidence supporting the hypothesis that cognitive need predicts individual innovative behaviors and can be considered a driving force behind individual innovation. Individuals with a high need for cognition tend to seek out and appreciate unique, complex, and uncertain situations, actively extracting information from their environment [66]. Moreover, individuals with a strong need for cognition demonstrate the ability to connect new and existing knowledge and acquire new information in a flexible and effective manner [67].

According to [57], an employee's relationship with his/her supervisor is a critical component of his/her immediate work environment as it influences his/her belief in the potential performance and image outcomes of his/her innovative efforts. According to the leader–member exchange (LMX) theory [68], workers who have strong relationships with their managers are rewarded with more resources, autonomy, and decision-making power in exchange for greater loyalty and dedication. A positive relationship with their supervisor can contribute to an employee's confidence in their innovative abilities and perceived support for their innovative endeavors.

Employee attitudes, beliefs, and behaviors are influenced by leaders' tolerance for and response to deviance [58]. Employee cognitions and behaviors can be significantly influenced by a leader's individual response to deviance [58]. Previous research on perceived deviance tolerance has emphasized its negative effects, such as a lack of moral engagement, deviant behavior, and aggressiveness [58]. The research conducted by the authors of [69] indicated that perceived deviance tolerance plays a positive role in encouraging innovative behaviors within the Chinese context. Their findings suggest that when employees perceive a higher level of tolerance for deviant behaviors in the workplace, it reduces their excessive fear of punishment and creates a more relaxed work environment. In such an environment, employees feel more comfortable expressing their ideas and taking initiative, which enhances their willingness to engage in innovative behaviors [70]. The perception of deviance tolerance can, therefore, contribute to creating a supportive atmosphere that fosters innovation within an organization.

## 4. Research Methodology

A three-phase research methodology was used in this study to examine and select EISB factors in the manufacturing industry that have the most impact on employee innovative behaviors from the perspective of managers. This research study was based on the work described in [71].

To identify the factors impacting Employee Social Intrapreneurial Behavior (ESIB) within the organization, a manager's perspective examination was conducted. These factors were categorized into two distinct groups: external and internal. This study relies on the perspectives of business professionals and utilizes the Analytic Hierarchy Process (AHP), a well-established approach for consolidating expert decisions and producing reliable results. The AHP is widely used in multi-criteria decision making (MCDM) and is considered one of the most practical strategies [72]. The AHP can be classified as an analytical and deductive method, operating on the basis of eliminating less important criteria or alternatives that are less likely to happen.

The AHP enables the transformation of qualitative judgments into quantitative values. A questionnaire that received eleven responses from professionals in the manufacturing industry, validating the suitability of the AHP for this research, was distributed.

The research approach consisted of three phases. In Phase I, an extensive literature review was conducted to identify the criteria and sub-criteria related to Employee Social Intrapreneurial Behavior (ESIB) factors and relevant initiatives. Following this, an expert panel (EP) was formed to contribute to the determination of the ESIB criteria and sub-criteria through thorough discussions and collaboration. In Phase II, the relative weights of the ESIB factors were analyzed using well-established Analytic Hierarchy Process (AHP) methods. A questionnaire was distributed to industry professionals to gather their perspectives and insights. The responses obtained were analyzed to determine the linguistic variable weights of the ESIB factors. Finally, Phase III involved ranking the different ESIB factors based on the linguistic variable weights obtained in Phase II. This ranking process provided a comparative assessment of the different ESIB factors. Figure 1 presents a schematic diagram illustrating the research approach employed in this study.

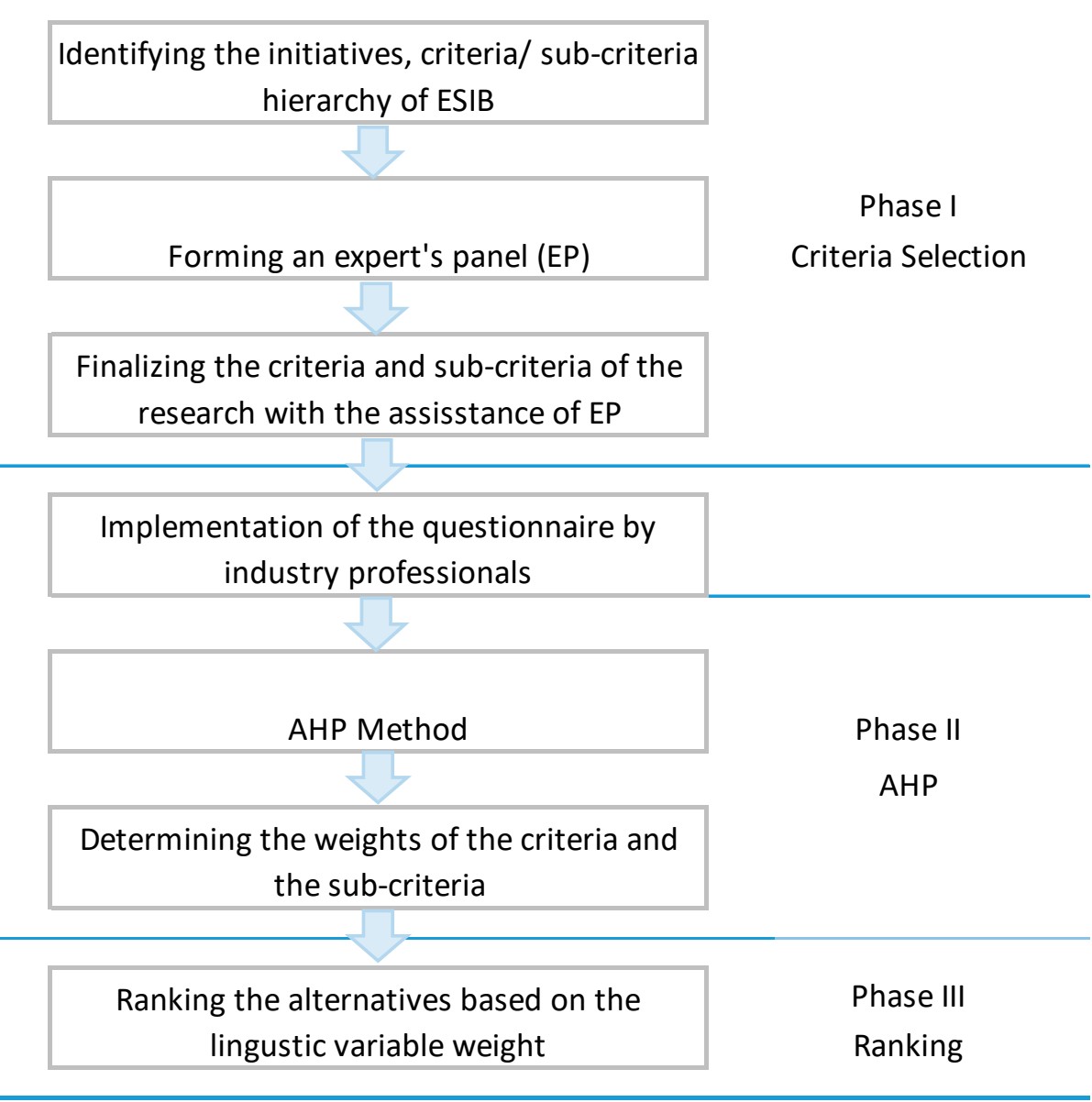

**Figure 1.** Three phase research methodology for ESIB programs.

### 4.1. Phase I: AHP

Determining the relevant criteria is an essential step in MCDM settings. Coaction during the selection process involved a combination of reviewing the existing literature and analyzing industrial experience. The present study applied two key criteria for selecting ESIB factors: "External organizational context of ESIB" and "Internal organizational context of ESIB". The hierarchy within these two criteria was determined in accordance with the classification provided by [23–27,54–58].

The EP used a 5-point Likert-scale questionnaire to screen and evaluate the criteria collected from the existing literature. The selection of the criteria was based on their average scores, and a final set of criteria were chosen. In this study, the criteria, sub-criteria, and alternatives were carefully selected through a comprehensive examination of the literature and current industry practices, with a specific emphasis on the manufacturing industry. Table 1 presents the outcomes of the identified criteria, sub-criteria, and relevant initiatives related to Employee Social Intrapreneurial Behavior (ESIB).

**Table 1.** ESIB initiatives in Manufacturing Industry.

| Criteria | Sub Criteria | Initiatives | Source |
|---|---|---|---|
| External | Search breadth | The number of External innovation partners as: customers, suppliers, competitors, universities, knowledge institutions. | [23] |
| | Collectivism culture | A manager who encourages loyalty and a sense of duty in subordinates than it is to encourage individual initiative. | [24] |
| | Dynamic work environment | A dynamic work environment/work environment provides opportunities for change. | [25] |
| | Corporate reputation | The company supports good causes that benefits society and environment/The company is good to work, both for its infrastructure, as for the work environment, benefits, and good practices with its employees. | [26] |
| | Culture intelligence | Informing employees about the cultural values and religious beliefs of other cultures/training employees to be confident & help them to socialize with locals in a culture that is unfamiliar to them/train employees for the cultural knowledge while they interact with people from a culture that is unfamiliar to them. | [27] |
| Internal | Perceived organization support | Encouraging creativity, allow people to find different ways to solve same problems, open organization & responsive to change. | [54] |
| | Expected imagine gains | Allowing employees to suggest, to participate in implementation of new ideas/allowing them to search out new techniques. | [55] |
| | Need for cognition | Giving responsibility to employees in the thinking process | [56] |
| | Superior relationship quality | Understanding employees 'job problems and needs/offering to employees help to solve problems at work. | [57] |
| | Perceived deviance tolerance | Tolerate employees' behaviors that damage the property of the company, employees' behaviors that lower the performance/employees' behaviors that cause personal attack on other workmates. | [58] |

### 4.2. Phase II: AHP

The qualitative aspect of evaluating ESIB factors requires a multi-criteria perspective. A widely preferred methodology in such contexts is the Analytic Hierarchy Process (AHP), developed by [73]. The AHP organizes decision components into a structured hierarchy, allowing a combination of subjective judgments and objective assessments to determine the best alternative. It systematically integrates qualitative and quantitative analyses, thereby enhancing the reliability of the decision-making process. Figure 2

illustrates the hierarchical structure of the AHP, consisting of the goal, decision criteria, and options for ESIB factor evaluation.

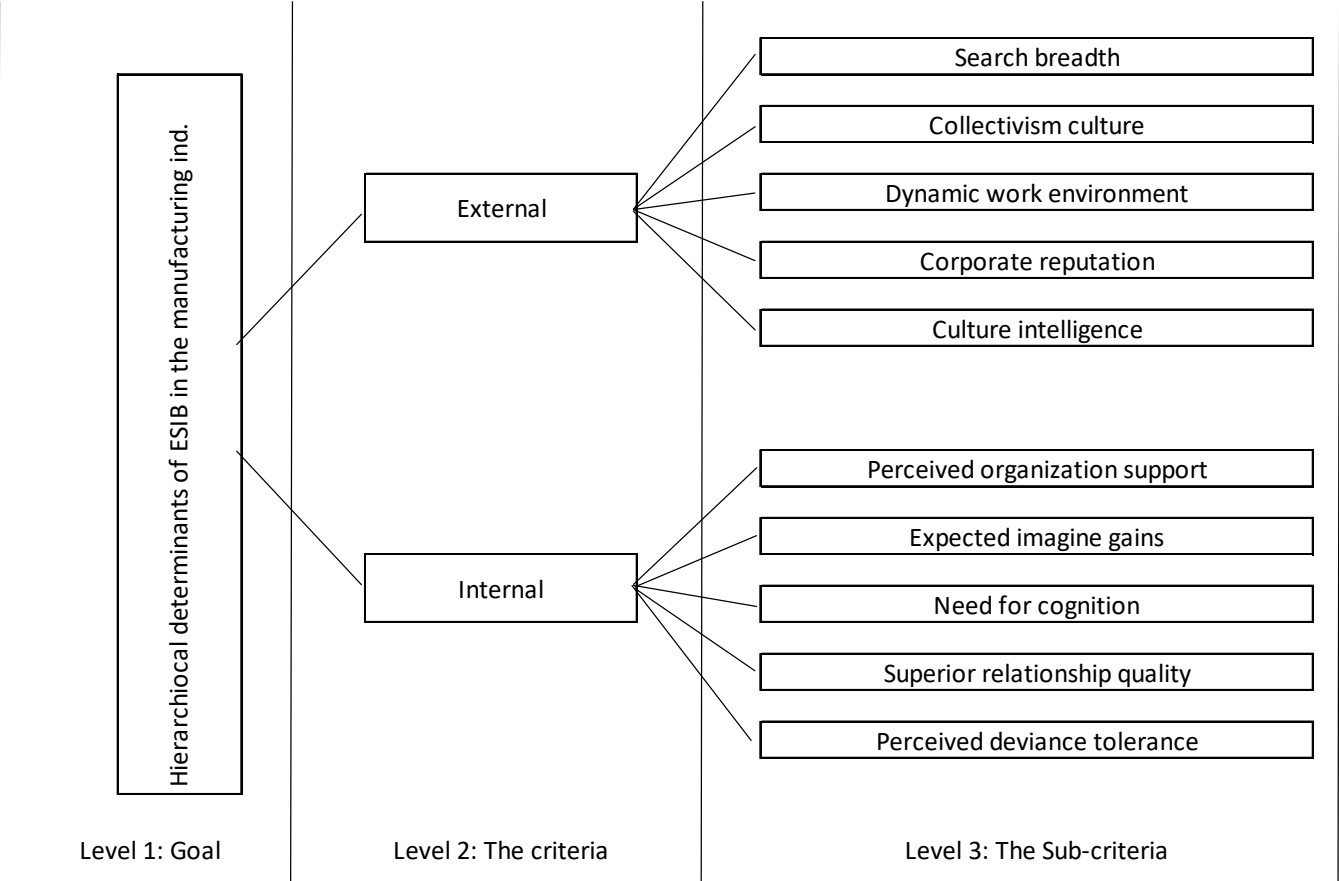

**Figure 2.** Hierarchy of the AHP model.

The next step is to compare the criteria and the choices pairwise. The relative importance of the criteria within each level and their alternatives is determined via prioritization. As demonstrated in Table 2, the pairwise comparisons were based on a standardized measure with nine levels. The pairwise comparison of n criteria is summarized in a nxn pairwise assessment matrix. The Thomas Saati scale Analytic Hierarchy Process (AHP) is a decision-making procedure originally developed by Thomas Saaty [73–75].

**Table 2.** Scale for pairwise comparisons.

| Intensity of Importance | Definition |
|---|---|
| 1 | Equal Importance |
| 3 | Moderate Importance |
| 5 | Strong Importance |
| 7 | Very Strong Importance |
| 9 | Absolute Importance |

2, 4, 6, 8 can be used to express intermediate values.

Let us define C = Cj j = 1; 2;/:n: as a set of ESIB factor criteria. A comparison of the criteria from the set C is included in the nxn evaluation matrix. Equation (1) provides the matrix *A*:

$$A = \begin{vmatrix} a_{11} & a_{12} \ldots & a_{1n} \\ a_{21} & a_{22} \ldots & a_{2n} \\ a_{n1} & a_{n2} \ldots & a_{nm} \end{vmatrix} \qquad (1)$$

Based on the AHP hierarchy shown in Figure 2, aij indicates the numerical assessment of the pairwise comparison between criteria i and j in this context. For example, if the condition i has absolute priority over the criteria j, then aij = 7, otherwise aij = 1/7. The entries of matrix *A* are determined by the following rules:

$$aij \neq 0, \; aii = 1, \; aji = \frac{1}{aij}$$

In the subsequent step of the Analytic Hierarchy Process (AHP) method, matrix *A* undergoes normalization and weight calculation. This is accomplished by dividing the entries in each column of matrix *A* by the corresponding column sums. The priority or precedence of the elements is then determined by the major eigenvector *w*, which corresponds to the maximum eigenvalue, $\lambda$max, of matrix *A*:

$$Aw = \lambda maxw$$

The final step in the AHP is to conduct a consistency analysis because the prominence of the AHP results is strongly dependent on the congruity of the pairwise comparison judgments. The consistency analysis is completed in two parts. To begin, the consistency index (*CI*) is calculated as follows:

$$CI = \frac{\lambda\text{max} - n}{n - 1}$$

Then, the final consistency ratio (*CR*) is calculated as follows:

$$CR = \frac{CI}{RI}$$

The use of *CR*, according to [71], is very important because it reveals the consistency of the paired assessments. The existing literature accepts 0.1 as an upper bound for *CR*.

*4.3. Phase III: Ranking*

Ranking the ESIB factors requires the application of the weights obtained in Phase II as well as expert opinions on the available alternatives. This is discussed in further depth in the following section.

## 5. The AHP Application in the Manufacturing Industry in Albania

A questionnaire was designed by the authors to measure pairwise relative weights in the Analytic Hierarchy Process (AHP) based on the hierarchical structure (See Appendix A). The questionnaire utilized a standardized AHP scale ranging from 1 to 9, with 1 representing neutral importance and 9 indicating absolute importance (Table 2). Judgmental sampling was employed to select survey participants, considering their expertise and involvement in Employee Social Intrapreneurial Behavior (ESIB) activities. Fifteen managers from four large manufacturing companies (Everest shpk., Lufra shpk., Ajka shpk., and F&M shpk.) in Albania were contacted, and their ratings on the relevance of various criteria and sub-criteria were collected using the questionnaire. The respondents were selected from managerial-level individuals who had participated in ESIB activities within their organizations. A total of eleven responses were received, resulting in a response rate of 73.3%. The collected questionnaire responses were analyzed using the well-known multi-criteria decision-making software Expert Choice, which handles standard AHP calculations. The local weights of the main criteria are presented in Table 3. As shown in the table, the internal sub-criteria account for up to 83% in terms of importance in determining employee social innovative behaviors in Albanian manufacturing sector.

**Table 3.** AHP weights and rank of the main criteria.

| Criteria | Weights | Rank |
|---|---|---|
| External | 0.172793 | 2 |
| Internal | 0.827207 | 1 |

As shown in the above table, the sub-components of the internal factor, such as superior relationship quality and expected image gains, have a combined value of 50% in terms of importance in the local weights for the Albanian context. Local weights represent the individual importance of each sub-criterion among the internal factors that impact ESIB. Considering the combined influence of the internal sub-criteria, their respective sub-factor global weights were calculated as well, and the results are presented in Table 4. The two abovementioned sub-components remain significant, accounting for 41% of global influence. These results agree with the findings of [76] on Albanian culture, where the power distance dimension ranks first, accounting for 90% and, thus, suggesting a society that is highly hierarchical. In such an environment, people accept and try to please their leaders. Furthermore, high uncertainty avoidance determines individual motivation for financial gains in an unsecure economic environment. Cultural intelligence has the highest influence, accounting for 30% in terms of local weight, although its global weight remains insignificant with a value of 5%. Again, uncertainty avoidance determines Albanian individuals' behavior to navigate and understand different cultures, which can contribute to innovative behavior.

**Table 4.** The local weights of the sub-criteria and their corresponding global weights. Local weights are determined at the second level of the hierarchy, and they are multiplied by the weight of the main criteria (first level) to calculate the global weights.

| AHP Weights and Rank of the Sub-Criteria | | | | | |
|---|---|---|---|---|---|
| Criteria | Sub-Criteria | Local Weights | Local Rank | Global Weights | Global Rank |
| External | Search breadth | 0.081823473 | 5 | 0.0141386 | 5 |
| | Collectivism culture | 0.197953751 | 3 | 0.0342051 | 3 |
| | Dynamic work environment | 0.237205276 | 2 | 0.0409875 | 2 |
| | Corporate reputation | 0.191439913 | 4 | 0.0330796 | 4 |
| | Culture intelligence | 0.291577586 | 1 | 0.0503827 | 1 |
| Internal | Perceived organization support | 0.132431983 | 4 | 0.1095486 | 4 |
| | Expected imagine gains | 0.247124175 | 2 | 0.2044227 | 2 |
| | Need for cognition | 0.202844045 | 3 | 0.1677939 | 3 |
| | Superior relationship quality | 0.259741203 | 1 | 0.2148596 | 1 |
| | Perceived deviance tolerance | 0.178750151 | 5 | 0.1478633 | 5 |

Subjective personal judgments during the AHP approach might lead to inconsistencies, reducing the reliability of the conclusions. As a result, a consistency ratio (CR) is computed as a measure of evaluation coherence [71]. A 0.1 upper limit is commonly accepted in the literature. If the consistency ratio exceeds the upper limit, the evaluation method must be repeated. In this research, the consistency check was applied to assess the degree of coherence among expert perspectives.

It was confirmed that consistency ratio is always below the critical value at all levels; The CR of the main criteria is by default 0.0, the CR of the external sub-criteria is 0.068, and the CR of the internal sub-criteria is 0.086, which are all below the threshold value, demonstrating the robustness of the results.

Given the fuzzy nature of the ESIB factor-ranking method, decision makers are more confident in providing interval value judgments rather than fixed value evaluations. The use of linguistic variables helps accommodate imprecise preferences and enhance the decision-making process, compensating for "ill-conditioned" quantitative descriptions. Linguistic variables reflect the vagueness of human logic: rather than expressing things

in numbers, people prefer doing so in words [77]. In this study, linguistic variables based on [78] were employed. The author of [78] expressed linguistic variables whose states are fuzzy numbers. He assigned the means of these fuzzy numbers as their look-up values. Table 5 presents the linguistic values, fuzzy numbers, and corresponding scale values utilized in the analysis.

**Table 5.** Values and scale values.

| Linguistic Valus (LV) | Fuzzy Numbers | Scale Value (SV) |
|---|---|---|
| Very good (VG) | (0.75, 1.00,1.00) | 1 |
| Good (G) | (0.50, 0.75, 1.00) | 0.75 |
| Fair (F) | (0.25, 0.50, 0.75) | 0.5 |
| Poor (P) | (0, 0.25, 0.50) | 0.25 |
| Very poor (VP) | (0, 0, 0.25) | 0 |

Two ESIB programs were chosen for assessment, and the results are presented in Table 6. The assessment aimed to show whether one program has an advantage over the other program by determining which achieves a higher score. The global weights (GWs) calculated using the AHP approach were used to rank the ESIB programs. These two programs were proposed as programs that organizations may implement to foster and motivate employees to participate in social innovative behaviors.

**Table 6.** Ranking of alternative ESIB programs.

| Sub-Criteria | GW | Workshop | | | Regional Clustering | | |
|---|---|---|---|---|---|---|---|
| | | LV | SV | GW × SV | LV | SV | GW × SV |
| Search breadth | 0.014139 | VP | 0 | 0 | VG | 1 | 0.0141386 |
| Collectivism culture | 0.034205 | F | 0.5 | 0.0171026 | G | 0.75 | 0.0256538 |
| Dynamic work environment | 0.040988 | G | 0.75 | 0.0307406 | VG | 1 | 0.0409875 |
| Corporate reputation | 0.03308 | VG | 1 | 0.0330796 | VG | 1 | 0.0330796 |
| Culture intelligence | 0.050383 | G | 0.75 | 0.037787 | F | 0.5 | 0.0251913 |
| | | | | | | | 0 |
| Perceived organization support | 0.109549 | VG | 1 | 0.1095486 | G | 0.75 | 0.0821615 |
| Expected imagine gains | 0.204423 | VG | 1 | 0.2044227 | F | 0.5 | 0.1022114 |
| Need for cognition | 0.167794 | G | 0.75 | 0.1258454 | F | 0.5 | 0.083897 |
| Superior relationship quality | 0.21486 | F | 0.5 | 0.1074298 | F | 0.5 | 0.1074298 |
| Perceived deviance tolerance | 0.147863 | VP | 0 | 0 | VP | 0 | 0 |
| | | | | 0.66596 | | | 0.51475 |

## 5.1. The First Program: "Workshop"

This program's main aim is to foster a positive organizational climate. In [79], humor in the workplace is defined as employee experiences and practices with good and negative types of humor during social interactions at work. The numerous positive outcomes of workplace humor can be classified into three categories: health promotion, mental flexibility enhancement, and social interaction improvement [80]. Previous research has examined the effect of humor in driving employee innovation [81]. In research on the concept exploration and idea generation stages [82], humor has been found to improve creative thinking [83] and openness to new ideas [84]. According to [85], a positive attitude promotes mental flexibility, which encourages the creation of original ideas and participation in creative endeavors. Additionally, humor offers a new perspective on job duties and helps connect ideas that at first glance may appear unconnected [86]. For instance, [86] discovered that

participating in improvisational comedy workshops had the potential to increase idea generation output by over 30% for engineers and product developers. Building on this research, this study proposed that implementing monthly workshops focusing on creating humorous situations could have a positive impact on Employee Social Intrapreneurial Behavior (ESIB).

*5.2. The Second Program: "Regional Clustering"*

In knowledge-intensive environments, fostering employees' intrapreneurial behavior is crucial for organizational innovation. Recent research has examined how specific human resource management (HRM) practices implemented in a firm can stimulate intrapreneurial behavior and innovation [87]. Employees are encouraged to seek new internal or external opportunities to enhance their company's performance and gain a competitive advantage [13].

Clusters, as emphasized by [88], have been identified as key determinants of regional competitiveness. They facilitate technological change, knowledge development, and innovation within member companies [89]. Clusters also reduce coordination costs, enable specialization in the workforce, and promote knowledge creation.

Research has shown that HRM practices in cluster environments impact organizational innovation and knowledge exchange [90]. Clusters attract and retain talent, providing access to valuable knowledge from consumers, suppliers, institutions, and competitors [91]. Organizations within clusters adapt their HRM practices to effectively utilize this information, thus fostering innovation and intrapreneurial behavior. Consequently, it is argued that participation in regional clusters can have positive impacts on HRM practices, leading to enhanced Employee Social Intrapreneurial Behavior (ESIB).

The expert panel utilized the global weights and linguistic values presented in Table 5 to rank the programs. The linguistic expressions were converted into scale values, and the panel provided their opinions, using terms such as very good, good, fair, poor, and very poor. These opinions were then translated into fuzzy numbers using established methods [71,78]. The precise calculations are presented in Table 6, revealing that ESIB 1 achieved the highest ranking, followed by ESIB 2.

After considering all sub-criteria, the experts strongly suggested implementing the practices of the Workshop instead of Regional Clustering. Combined with their global weights, the total influence of internal sub-criteria for company activities is 0.66, which is higher than the influence of external sub-criteria for company activities of 0.51. Once again, these findings emphasize a higher importance of internal organizational context in determining ESIB for the Albanian manufacturing sector.

## 6. Discussion

The findings of this study indicate that, when compared to the exterior (0.172) dimensions, the "internal" sub-criteria receive a substantially higher overall score (0.827). Cultural intelligence and dynamic work environment receive the highest ranking under the "External" component when compared to collectivism culture, corporate reputation, and search breadth. According to [37], employees perform more innovatively when they operate in a dynamic workplace. Employees with a higher cultural intelligence level are more motivated to connect often and productively with coworkers from other cultural backgrounds, which may boost their social context centrality and enable them to learn from others [50]. As a result, these workers are more likely to engage in creative jobs, exert significant effort to accomplish difficult goals, and develop and implement innovative ideas even in trying situations [52]. In a fast-paced workplace, having exceptional workers is essential for fostering organizational success. In the Albanian context, employees have a high level of adaptability to working with diverse working groups and do not fear collaboration and communication with others; this may be attributed to their abilities to learn new languages and immigration from Albania in the early 1990s.

Working in a dynamic work environment may impact Albanian employees because they are more ready to work in such an environment than in a stable work environment.

According to the findings, internal factors are considered as the main drivers of ESIB. It is interesting to note that under the "Internal" dimensions, factors related to superior relationship quality and expected image gains take priority over other factors relating to the need for cognition, perceived organizational support, and perceived deviance tolerance. According to the leader–member exchange (LMX) theory [68], employees who have a good relationship with their supervisors are given more resources, decision latitude, and freedom in exchange for greater commitment and loyalty.

Another aspect to consider is the style of leadership, which can have an impact on the relationship quality between leaders and their followers. According to [92], leaders who demonstrate both opening leadership behaviors (encouraging followers to do things differently) and closing leadership behaviors (including corrective action, routines, and sanctioning mistakes) can contribute to the development of innovative work behaviors in their followers. This research came to similar results. The main inspiration that might push employees into engaging in innovative but socially oriented behaviors in the Albanian manufacturing industry is leaders and their style of leading the organization. In the Albanian context, leaders can foster the potential of their followers, and the bond that leaders build with their followers has an impact on employees' behaviors and can encourage them to engage in social innovative behaviors. In contrast to the authors of [55], who claimed that expected image gains have a significant negative effect on innovative behaviors, the findings of this study showed that the impact appears to be positive according to the experts who participated in the research. In the Albanian context, one inspiration for employees to engage in activities that demand more effort and time is the perception of expected gains such as bonuses, promotions, or payment increases. Albania is one of the Western Balkan countries with unstable economic and political conditions. The need for having a stable life is crucial, and many young people are migrating to other European countries because of the insufficient benefits that they receive in their jobs. According to the experts who participated in this study, if employees expect an increase in financial benefits, they will engage in social intrapreneurial initiatives and contribute to their organization.

We believe that by focusing on employees' social innovative behaviors in the Albanian manufacturing industry, our research fills an important gap in the literature. HR and managers can gain an understanding of the factors that may encourage workers to contribute more within their organization to nurture innovative behaviors. The development of an organization relies on employees who are at the bottom of the pyramid. Employees are a vital asset to organizations, and their contributions should be recognized.

In this study, we employed AHP methodology to analyze factors influencing ESIB. In the first step of developing the model, a panel of experts examined the interdependence of the criterion and sub-criteria. This research aimed to evaluate the hierarchical factors of ESIB, which involved distinct components related to internal and external criteria. Naturally, different criteria attracted differing levels of attention from responders; measuring their relevance levels required a thorough separation among the concepts. The relationship between various ESIB criteria and sub-criteria and the expert assessment show different levels of importance. The criteria and sub-criteria exhibit no significant interdependence. This means that the influence of one criterion on another is either non-existent or minimal (i.e., the level of influence is not significant). Additionally, it was determined that there is no interdependence between the levels.

## 7. Conclusions

This study has limitations that can be addressed in future research. Although the proposed model can be applied worldwide, the viewpoints on ESIB were obtained solely from manufacturing experts in Albania, which might be considered a limitation. As a result, regional applications should be carried out to foster innovative behaviors

inside businesses if the manufacturing industry is interested in leveraging scientific models like the one developed in this study. The proposed model can be applied with variable degrees of success around the globe. Comparing certain pairs or groups of ESIB applications to see which one is more advantageous could be an extension of this research study. By comparing different applications, it could raise awareness of areas for improvement in ESIB development.

The survey responses were obtained from manufacturing professionals, and perspectives from other stakeholders such as employees and external organizations were not included in the study. Future research could explore the viewpoints of employees and other stakeholders like NGOs or legal institutions. Additionally, this study focused on large and economically stable manufacturing companies. Conducting similar research with a larger and more diverse sample, including organizations of different sizes and from different sectors, would be beneficial. Furthermore, an alternative multi-criteria approach like the ANP could be employed in a similar context. Furthermore, as part of future research, innovative organizational culture might be included as one of the antecedents, as [93] determined that companies should develop organizational culture, boost leadership–member communication, and create an enjoyable work environment.

This study offers a comprehensive and practical MCDM model to address the challenge of selecting ESIB initiatives in the manufacturing sector. It focuses specifically on the Albanian manufacturing industry, which plays a crucial role in the overall development, productivity, and competitiveness of the industry. This study fills a gap in the literature by examining ESIB in the context of manufacturing in Albania, highlighting the tendency to prioritize internal aspects over external dimensions. The research findings provide valuable insights into ESIB practices in Albanian manufacturing companies. This study utilizes the AHP approach to prioritize the criteria for ESIB programs, offering a robust and flexible methodology.

**Author Contributions:** Methodology, N.B.; Validation, S.Z.; Writing—original draft, A.M.; Writing—review & editing, M.T. All authors have read and agreed to the published version of the manuscript.

**Funding:** This research received no external funding.

**Institutional Review Board Statement:** Not applicable.

**Informed Consent Statement:** Informed consent was obtained from all subjects involved in the study.

**Data Availability Statement:** Not applicable.

**Conflicts of Interest:** The authors declare no conflict of interest.

## Appendix A

| Employee Social Inovativeness Behaviour in Manufacturing Industry |
|---|

**Instructions:**

Please compare the below criteria / subcriteria couplet based on their relative importance to each other.

Only one box is going to be checked in the comparisons.

If the item on the left is more important than the item on the right, please use the scale on the left of 1 and indicate its relative importance.

If the item on the right is more important than the item on the left, please use the scale on the right and indicate its importance.

If the items have equal importance, than color the box for 1.

| Intensity of Importance | Definition |
|---|---|
| 1 | Equal Importance |
| 3 | Moderate Importance |
| 5 | Strong Importance |
| 7 | Very Strong Importance |
| 9 | Absolute Importance |
| 2, 4, 6, 8 can be used to express intermediate values. | |

| COMPARING THE MAIN CRITERIA | | | | | | | | | | | | | | | | | | |
|---|---|---|---|---|---|---|---|---|---|---|---|---|---|---|---|---|---|---|
| | 9 | 8 | 7 | 6 | 5 | 4 | 3 | 2 | 1 | 2 | 3 | 4 | 5 | 6 | 7 | 8 | 9 | |
| External | | | | | | | | | | | | | | | | | | Internal |

| COMPARING THE SUBCRITERIA | | | | | | | | | | | | | | | | | | | |
|---|---|---|---|---|---|---|---|---|---|---|---|---|---|---|---|---|---|---|---|
| | | 9 | 8 | 7 | 6 | 5 | 4 | 3 | 2 | 1 | 2 | 3 | 4 | 5 | 6 | 7 | 8 | 9 | |
| External | Search breadth | | | | | | | | | | | | | | | | | | Collectivism culture |
| | Search breadth | | | | | | | | | | | | | | | | | | Dynamic work environment |
| | Search breadth | | | | | | | | | | | | | | | | | | Corporate reputation |
| | Search breadth | | | | | | | | | | | | | | | | | | Culture intelligence |
| | Collectivism culture | | | | | | | | | | | | | | | | | | Dynamic work environment |
| | Collectivism culture | | | | | | | | | | | | | | | | | | Corporate reputation |
| | Collectivism culture | | | | | | | | | | | | | | | | | | Culture intelligence |
| | Dynamic work environment | | | | | | | | | | | | | | | | | | Corporate reputation |
| | Dynamic work environment | | | | | | | | | | | | | | | | | | Culture intelligence |
| | Corporate reputation | | | | | | | | | | | | | | | | | | Culture intelligence |
| | | 9 | 8 | 7 | 6 | 5 | 4 | 3 | 2 | 1 | 2 | 3 | 4 | 5 | 6 | 7 | 8 | 9 | |
| Internal | Perceived organization support | | | | | | | | | | | | | | | | | | Expected imagine gains |
| | Perceived organization support | | | | | | | | | | | | | | | | | | Need for cognition |
| | Perceived organization support | | | | | | | | | | | | | | | | | | Superior relationship quality |
| | Perceived organization support | | | | | | | | | | | | | | | | | | Perceived deviance tolerance |
| | Expected imagine gains | | | | | | | | | | | | | | | | | | Need for cognition |
| | Expected imagine gains | | | | | | | | | | | | | | | | | | Superior relationship quality |
| | Expected imagine gains | | | | | | | | | | | | | | | | | | Perceived deviance tolerance |
| | Need for cognition | | | | | | | | | | | | | | | | | | Superior relationship quality |
| | Need for cognition | | | | | | | | | | | | | | | | | | Perceived deviance tolerance |
| | Superior relationship quality | | | | | | | | | | | | | | | | | | Perceived deviance tolerance |

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
