# Peer review of "ESIB’s Antecedents: An Analytic Hierarchy Process Application in the Manufacturing Industry in Albania"

_sustainability, doi:10.3390/su151813838_

Round 1

Reviewer 1 Report

I appreciate the opportunity to explore such interesting research.

The article's strengths are its relevance, logical structure and an interesting idea as the basis of the article. The arguments and discussion of findings are coherent, balanced and compelling. Yet, despite the merits, the presentation of the study could be better in the following areas:

  1. Indicate the purpose of the study clearly, now, it is a little blurry. Perhaps it will be more straightforward for the reader If you indicate hypotheses or research questions.
  2. There is a significant gap in the analysis of the literature, especially in the newest one (2023), concerning "corporate social responsibility", it can be useful "Dudek M. Methodology for assessment of inclusive social responsibility of the energy industry enterprises". "Malynovska Y. Enhancing the Activity of Employees of the Communication Department of an Energy Sector Company" – maybe this one would be useful concerning employees.
  3. I cannot find a description of how many enterprises participated in the methodology testing. In principle, this part needs to be described in more detail.
  4. Mention the article's limitations in a separate section.
  5. In the Discussion section point more to the comparison with existing studies, in which it is necessary to emphasize the novelty of the study and its practical/theoretical/methodological contribution and highlight how the author's research differs from them.

Author Response

Response to Reviewer 1 Comments

Point 1: Are all the cited references relevant to the research?

Response 1: Has been reflected by eliminating some references. (in red)

[74] Akman, E., & Karaman, A. (2021). Pillars in the making, Industry 4.0 on the horizon. International Journal of the Analytic Hierarchy Process, 13(2), 240-277.

Point 2: For empirical research, are the results clearly presented?

Response 2: Has been reflected by including a more detailed explanation on page no. 10-11-12.

Point 3: Is the article adequately referenced?

Response 3: It has been added the references as below:

[21] Dudek, M., Bashynska, I., Filyppova, S., Yermak, S., & Cichoń, D. Methodology for assessment of inclusive social responsibility of the energy industry enterprises. Journal of Cleaner Production. (2023). 136317. URL: https://doi.org/10.1016/j.jclepro.2023.136317

[75] Country comparison tool. (n.d.). https://www.hofstede-insights.com/country-comparison-tool?countries=albania*

Point 4: Indicate the purpose of the study clearly.

Response 4: Has been reflected by including a more detailed explanation on page no. 2(in red)

 Point 5: I cannot find a description of how many enterprises participated in the methodology testing. In principle, this part needs to be described in more detail.

Response 5:  Has been reflected by including a more detailed explanation on page no. 10(in red)

 Point 6: Mention the article's limitations in a separate section.

Response 6:  Has been reflected by including a more detailed explanation on page no. 15(in red)

Point 7: In the Discussion section point more to the comparison with existing studies, in which it is necessary to emphasize the novelty of the study and its practical/theoretical/methodological contribution and highlight how the author's research differs from them.

Response 7: Has been reflected by including a more detailed explanation on page no. 14(in red)

Reviewer 2 Report

Try to simplify the title of an article by using an 8 to 15 words rule of thumb. 

Your abstract needs to be explained sufficiently. Rewrite by shifting the definition of ESIB in the literature review and following the guidelines: i.e. The abstract should start with objectives. Then the method is applied to carry out the investigation and gives us brief research findings (and, in conclusion, their implications).

The keywords will be revised. Try to add those keywords that are not part of the title or abstract.

The introduction shall follow a classical approach: starting by defining dependent variables or problems, then brief literature in the second paragraph. The third paragraph of the intro shall give us gaps and significance, and lastly, your key contributions (method, theory, etc.), followed by research objectives or research questions.

Your article shall add a theoretical underpinning section to the literature review section and let us know the key theory linked to the literature review, e.g. Schumpeter's theory of innovation or any other base theory. Read the following article and cite the relevant theory and also cite:

Saleem, I., Siddique, I., & Ahmed, A. (2019). An extension of the socioemotional wealth perspective: Insights from an Asian sample. Journal of Family Business Management10(4), 293-312.

Try to enhance your discussion by expanding on your contributions from the intro, then add future research, compare with old studies, and limitations of your method etc.

Cite the following articles for future research:

Kumar, S., Raj, R., Saleem, I., Singh, E. P., Goel, K., & Bhatia, R. (2023). The interplay of organisational culture, transformational leadership and organisation innovativeness: Evidence from India. Asian Business & Management, 1-31.

Author Response

Response to Reviewer 2 Comments (in blue)

Point 1: Try to simplify the title of an article by using an 8 to 15 words rule of thumb.

Response 1: Has been reflected by changing the title. on page no.1. (In blue)

Point 2: Your abstract needs to be explained sufficiently. The keywords will be revised.

Response 2: Has been reflected by including a more detailed explanation on page no.1.

Point 3: The introduction shall follow a classical approach: starting by defining dependent variables or problems, then brief literature in the second paragraph. The third paragraph of the intro shall give us gaps and significance, and lastly, your key contributions (method, theory, etc.), followed by research objectives or research questions.

Response 3:  Has been reflected by including a more detailed explanation on page no.2-3

Point 4: Your article shall add a theoretical underpinning section to the literature review section and let us know the key theory linked to the literature review, e.g. Schumpeter's theory of innovation or any other base theory. 

Response 4: Has been reflected by including a more detailed explanation on page no.3

 Point 5: Try to enhance your discussion by expanding on your contributions from the intro, then add future research, compare with old studies, and limitations of your method etc.

Cite the following articles for future research: Kumar, S., Raj, R., Saleem, I., Singh, E. P., Goel, K., & Bhatia, R. (2023). The interplay of organisational culture, transformational leadership, and organisation innovativeness: Evidence from India. Asian Business & Management, 1-31.

 Response 5:  Has been reflected by including a more detailed explanation on page no.17

Reviewer 3 Report

·        Elaborate AHP in the title

·        It is unusual for an abstract to start with a quote or definition

·        Citation style is flawed in many places throughout the document (please refer to the journal guide)

·        Paragraph 3 on page 2 “ To identify ……………., was distributed” – the whole paragraph should go under the Methodology section

·        Standard of writing and sentence construction need to be significantly addressed. For example, the sentence, “ It’s believed that by introducing a new concept of innovative behavior, Employee Social Innovativeness Behaviour, it is considered a contribution to the literature to a new level.”  -- is awkward and confounded. There are so many sentences throughout the document, which must be fixed by a professional English Language Editor.

· The focus/objective of the paper is confusing too. I would ask authors to precisely state the objective.

·        Literature review needs to be critical. The authors should contextualize the ESIB and its factors as per the study objective. As well, they should argue on the relevance of such a review in the context of the manufacturing industry. In the present form, the whole review looks like a plain definition and description of the ESIB and its factors. I suggest the authors re-write the whole literature review under one heading and put their arguments more convincingly based on the thematic areas that the research deals with.   More specifically, the authors should establish the argument that the external and internal sub-criteria correspond to the ESIB in a manufacturing industry.  To this end, I think, Table 1 should be discussed in a narrative (running text form) in the literature section, which means the Table to be wiped out from the Methods section.

·        The methods section is found to be vague. The authors did not specify what kind of approach it followed -- whether it is exploratory, inductive, deductive, or other. Again, citation issues made the whole Methods section difficult to understand. Concerns arise: as to which work/established methods the paper is based on, what were the procedures in procuring relevant literature, what is expert panel/who are the experts, who are industry professionals and how many, why they were selected, and through what process etc.    

·        The authors should elaborate on how the said classifications helped the earlier studies (as mentioned [55],[25],[27], [28] [22],[56],[57],[67],[59]) to obtain ESIB data/information.

·        Is there any example of using a 5-point Likert scale for screening and evaluating criteria collected from existing literature?  

·        The scale of pair-wise comparison is difficult to justify in AHP determination in the manufacturing industry. The entire Phase -2 methods should be captured by Fig 2. The Table and subsequent equations made the methods for this criteria quite complex and confounded. I suggest the authors get rid of these.  

·        In section 5, it is not clear where this 9-point scale comes from; and I think the questionnaire attached is flawed in many respects.

·        Overall, the paper attempted to cover a large ground and a wide array of concepts and issues both related and unrelated to ESIB. It must be re-written in a much simplified, precise, and concise way.

·        The authors also pay attention so that the conclusion addresses the study objective and flows from discussions/results – in the present form, this is not the case.

As mentioned earlier, the article needs to be re-written first and then edited by a professional English Language editor.  

Author Response

Response to Reviewer 3 Comments

Point 1:   Elaborate AHP in the title.

Response 1: Has been reflected. (In green)

Point 2: It is unusual for an abstract to start with a quote or definition.

Response 2: Has been reflected by including a more detailed explanation on page no. 1 (in green)

Point 3: Citation style is flawed in many places throughout the document (please refer to the journal guide)

Response 3: References have been reviewed and re-organized throughout the material.

Point 4:      Paragraph 3 on page 2 “ To identify ……………., was distributed” – the whole paragraph should go under the Methodology section

Response 4: Has been reflected by including a more detailed explanation on page no. 7(in green)

 Point 5: Standard of writing and sentence construction need to be significantly addressed. For example, the sentence, “ It’s believed that by introducing a new concept of innovative behavior, Employee Social Innovativeness Behaviour, it is considered a contribution to the literature to a new level.”  -- is awkward and confounded. There are so many sentences throughout the document, which must be fixed by a professional English Language Editor.

Response 5:  Has been reflected by sending the paper for English editing. Certificate has been provided by MDPI. (In green)

 Point 6: The focus/objective of the paper is confusing too. I would ask authors to precisely state the objective.

Response 6:  Has been reflected by including a more detailed explanation on page no. 1(in green)

Point 7: Literature review needs to be critical. The authors should contextualize the ESIB and its factors as per the study objective. As well, they should argue on the relevance of such a review in the context of the manufacturing industry. In the present form, the whole review looks like a plain definition and description of the ESIB and its factors. I suggest the authors re-write the whole literature review under one heading and put their arguments more convincingly based on the thematic areas that the research deals with.   More specifically, the authors should establish the argument that the external and internal sub-criteria correspond to the ESIB in a manufacturing industry.  To this end, I think, Table 1 should be discussed in a narrative (running text form) in the literature section, which means the Table to be wiped out from the Methods section.

Response 7: Literature review has been re-organized and has been enriched (in green).

Point 8: The methods section is found to be vague. The authors did not specify what kind of approach it followed -- whether it is exploratory, inductive, deductive, or other. Again, citation issues made the whole Methods section difficult to understand. Concerns arise: as to which work/established methods the paper is based on, what were the procedures in procuring relevant literature, what is expert panel/who are the experts, who are industry professionals and how many, why they were selected, and through what process etc.   

Response 8: AHP is one of multicriteria decision making that can be classified as an analytical method.

Point 9: The authors should elaborate on how the said classifications helped the earlier studies (as mentioned [55],[25],[27], [28] [22],[56],[57],[67],[59]) to obtain ESIB data/information.

Response 9: Has been reflected by including a more detailed explanation on page no. 5,7(in green)

Point 10:  Is there any example of using a 5-point Likert scale for screening and evaluating criteria collected from existing literature?  

Response 10: There are no other examples.

Point 11:   The scale of pair-wise comparison is difficult to justify in AHP determination in the manufacturing industry. The entire Phase -2 methods should be captured by Fig 2. The Table and subsequent equations made the methods for this criterion quite complex and confounded. I suggest the authors get rid of these. 

Response 11: This is considered a very important part of AHP. No action has been taken.

Point 12:  In section 5, it is not clear where this 9-point scale comes from; and I think the questionnaire attached is flawed in many respects.

Response 12: Thomas Saati scale The Analytic Hierarchy Process (AHP) is a decision-making procedure originally developed by Thomas Saaty (Saaty 1977, 1980, 1986).

Point 13: Overall, the paper attempted to cover a large ground and a wide array of concepts and issues both related and unrelated to ESIB. It must be re-written in a much simplified, precise, and concise way.

Response 13: Actions have been taken throughout the material, reflecting as well other reviewers’ notes.

Point 14: The authors also pay attention so that the conclusion addresses the study objective and flows from discussions/results – in the present form, this is not the case.

Response 14: Objectives have been added and conclusions and finding have been refined to reflect them accordingly.

Round 2

Reviewer 2 Report

Good luck

Author Response

Thank you for your valuable comments. 

Reviewer 3 Report

The quality of the paper has improved considerably after the revision. However, I would recommend the authors separate the Discussion section from Findings, i.e., Findings and Discussions should not go together. I think section 6.0 should be 'Discussion' only as the Results/or Findings are already exhibited in 5.0.

Second, as I advised earlier there should not be a separate section for 'Limitation and Future Research'. Please include this section (7.0) within Conclusions - and in a brief form.       

The authors should go through the journal citation style again and see whether they complied referencing procedure thoroughly. I guess, they need to revise many of their statements upon corrections of referencing. I leave it to the editors.   

Author Response

Point 1: However, I would recommend the authors separate the Discussion section from Findings, i.e., Findings and Discussions should not go together. I think section 6.0 should be 'Discussion' only as the Results/or Findings are already exhibited in 5.0.

Response 1: Has been reflected. (In red)

Point 2: Second, as I advised earlier there should not be a separate section for 'Limitation and Future Research'. Please include this section (7.0) within Conclusions - and in a brief form.      

Response 2: Has been reflected. (In red)

Point 3: The authors should go through the journal citation style again and see whether they complied referencing procedure thoroughly.

Response 3: References have been reviewed and re-organized throughout the material and adapted to the journal citation style.

Thank you for your valuable comments.